# Firefighting Water Jet Trajectory Detection from Unmanned Aerial Vehicle Imagery Using Learnable Prompt Vectors

**DOI:** 10.3390/s24113553

**Published:** 2024-05-31

**Authors:** Hengyu Cheng, Jinsong Zhu, Sining Wang, Ke Yan, Haojie Wang

**Affiliations:** 1School of Mechanical and Electrical Engineering, China University of Mining and Technology, Xuzhou 221006, China; tb18050003b0@cumt.edu.cn (H.C.); tb23050013a41ld@cumt.edu.cn (S.W.); ts23050139p31@cumt.edu.cn (K.Y.); ts23050202p31ty@cumt.edu.cn (H.W.); 2China Academy of Safety Science and Technology, Beijing 100012, China; 3Shenzhen Research Institute of China University of Mining and Technology, Shenzhen 518057, China

**Keywords:** jet trajectory, UAV image, learnable prompt vectors

## Abstract

This research presents an innovative methodology aimed at monitoring jet trajectory during the jetting process using imagery captured by unmanned aerial vehicles (UAVs). This approach seamlessly integrates UAV imagery with an offline learnable prompt vector module (OPVM) to enhance trajectory monitoring accuracy and stability. By leveraging a high-resolution camera mounted on a UAV, image enhancement is proposed to solve the problem of geometric and photometric distortion in jet trajectory images, and the Faster R-CNN network is deployed to detect objects within the images and precisely identify the jet trajectory within the video stream. Subsequently, the offline learnable prompt vector module is incorporated to further refine trajectory predictions, thereby improving monitoring accuracy and stability. In particular, the offline learnable prompt vector module not only learns the visual characteristics of jet trajectory but also incorporates their textual features, thus adopting a bimodal approach to trajectory analysis. Additionally, OPVM is trained offline, thereby minimizing additional memory and computational resource requirements. Experimental findings underscore the method’s remarkable precision of 95.4% and efficiency in monitoring jet trajectory, thereby laying a solid foundation for advancements in trajectory detection and tracking. This methodology holds significant potential for application in firefighting systems and industrial processes, offering a robust framework to address dynamic trajectory monitoring challenges and augment computer vision capabilities in practical scenarios.

## 1. Introduction

Forest fires, or wildfires, have caused significant loss of life and property damage in recent years [1]. They are classified as natural, caused by phenomena like lightning and dry weather, or human-caused, resulting from activities such as cooking and negligence. Approximately 90% of wildfires are human-caused [2], and the emission of toxic gases from these fires affects both humans and wildlife [3]. The emphasis on public safety has driven the transformation of fire safety equipment towards automation and intelligence [4,5,6]. Urban areas are increasingly mandating automatic fire suppression systems, and firefighting vehicles are becoming more automated [7,8,9]. Hence, there is a growing emphasis on swiftly and precisely delivering firefighting water to the fire point, making it a focal point of research. Identifying and locating the fire source, as well as automatically controlling the water flow landing point, are pivotal components of the automatic fire suppression system during firefighting operations [10,11,12]. Delivering firefighting water swiftly and accurately is crucial, yet research on water flow trajectory remains limited. Real-time monitoring of water flow trajectory is essential for effective firefighting and reducing response times. Recent advancements in large-scale language-vision models and the integration of prompt learning into computer vision offer promising solutions to these challenges [13,14,15,16]. Unmanned aerial vehicles (UAVs) provide rapid imaging, extensive coverage, and enhanced safety, making them ideal for this application [17,18,19,20]. This study explores detecting jet trajectory using UAV imagery and computer vision models with learnable prompt vectors.

Recently, there has been a surge in deep learning-based fire detection research, driven by the exceptional performance of detection networks rooted in deep learning in tasks such as localization and classification. Chen [21] proposed AIENet, an All-in-one Image Enhancement Network, designed to address various image degradation issues encountered in UAV monitoring tasks. AIENet incorporated a novel multi-scale receptive field image enhancement block, demonstrating superior performance compared to state-of-the-art algorithms in both qualitative and quantitative evaluations on synthetic forest datasets. Zheng [22] proposed a real-time fire detection algorithm to achieve the initial extraction of flame and smoke features while greatly reducing the computational effort of the network structure. To achieve a breakthrough in both algorithm speed and accuracy, Zheng [23] designed a two-stage recognition method that combines the novel YOLO algorithm with Real-ESRGAN. Zheng [24] introduced a trainable matrix in the encoder to compute features, reducing computational burden, emphasizing key features, and shortening training time, while also enhancing the encoding block by iteratively updating high- and low-level features, thereby reducing feature computation and remaining compatible with any state-of-the-art transformer decoder. Additionally, to address the multi-scale nature of fires and diverse environmental complexities, further modifications were made to accommodate varying scales and complexities. Chao [25] used a color-guided anchoring strategy based on flame color characteristics and introduced a global information-guided fire detection approach, enhancing Faster R-CNN by regionally integrating convolutional neural network features. Li [26] introduced a normalization-based attention module in DETR’s feature extraction and employed multi-scale deformable attention in the encoder-decoder structure to expedite model training convergence. Huang [27] enhanced Deformable-DETR with the integration of Multi-Scale Context Contrastive Local Feature Module (MCCL) and Dense Pyramid Pooling Module (DPPM) in the feature extraction module and introduced an iterative bounding box fusion technique for accurate bounding boxes enclosing complete smoke objects.

Although these approaches had achieved considerable success in fire detection, their input data were limited to image data, which was relatively singular and not conducive to a better understanding and description of the data by the model. Moreover, when image data were missing or subjected to noise interference, the performance of the model tended to degrade, with lower robustness to data anomalies. Recently, the development of prompt learning has, to some extent, alleviated this issue. Radford [15] proposed a self-supervised learning approach that learns multimodal representations by simultaneously training text and image encoders, with wide-ranging potential applications. However, Zhou [28] introduced a learnable prompt mechanism CoOp to better adapt the powerful and generalized priors of vision-language models, such as CLIP, to downstream tasks. CoOp was the first study in the computer vision domain to explore learnable prompts based on CLIP. The learnable vectors in CoCoOp [29], also known as dynamic prompts, consider the information of each input, making them less sensitive to category shifts. Visual Prompt Engineering is a broad concept that involves using prompts to guide model learning and inference in computer vision tasks [30]. Our approach, termed Prompt Computer Vision (PCV), specifically focuses on the application of learnable prompts in UAV imagery to enhance the accuracy of firefighting water jet trajectory detection. Unlike general Visual Prompt Engineering methods, PCV leverages multimodal data inputs and is tailored for the complex and dynamic environments encountered in firefighting operations.

In this study, we introduced a novel application of Visual Prompt Engineering, which we term Prompt Computer Vision (PCV), for accurately detecting jet trajectory using data acquired from UAV sensors. PCV leverages a multimodal data input approach, combining textual and visual information to enhance trajectory recognition accuracy. Our goal is to address the challenges posed by harsh fire environments and potential data sampling damage by proposing a method that integrates UAV image characteristics with a technique based on offline modules to learn prompt vectors (OMPV) for jet trajectory detection. The PCV method enables the rapid acquisition of high-quality jet trajectory images through the flexibility, safety, and efficiency of UAVs. The subsequent sections of this paper are organized as follows: Section 2 details the PCV method, Section 3 presents experiments, Section 4 presents results and discussion, and Section 5 concludes our findings.

The major contributions of the paper can be summarized as follows:(1)Image enhancement techniques are used to significantly resolve the radiometric distortion and geometric distortion of the UAV-based jet trajectory images. Data augmentation, such as flipping and adjusting image brightness, hue, contrast, and saturation, can enhance image features to address distortion and geometric distortion.(2)An offline module of learnable prompt vectors is designed to improve classification scores for detected jet trajectories. The simplicity of jet trajectory leads to basic feature extraction, hindering accurate classification. Prompt vectors, as textual features, complement network inputs, thereby improving classification accuracy and enhancing detection reliability.(3)A method for detecting jet trajectory from UAV imagery using learnable prompt vectors is proposed. The approach has a significant advantage in locating jet trajectory in adverse environments and classifying jet trajectory in feature-scarce scenarios.

## 2. The Prompt Computer Vision Methods

The proposed method involves several preparatory steps, followed by the core components of the PCV method. The preparatory steps include UAV image capture and image preprocessing, while the core components are learnable prompt vector generation and the jet trajectory model. First, aerial images of jet streams are collected using UAVs, and data augmentation is applied to these images to enhance sample diversity and model robustness. Subsequently, prompt vectors for learning jet trajectory are generated to improve model generalization and acquire textual features of jet trajectory. Finally, the visually augmented features and learned textual features are input into a computer vision model for jet trajectory detection. All parts will be described in more detail in the following sections.

### 2.1. UAV Image Capture

Capturing high-quality images is crucial for the visual detection of jet trajectory. The UAV sensor device is specifically designed to capture jet trajectory images, as it is challenging to fully capture trajectory images during jetting. The UAV sensor device consists of a fuselage, propellers, and a visual sensor, with the visual sensor fixed on the UAV fuselage. During firefighting, the water jet landing point is far from the fire monitor, making it difficult to capture complete trajectory images at the fire monitor’s location. Simultaneously, controlling the UAV sensor device to fly above the fire and maintaining a certain distance allows for the safe and efficient collection of jet trajectory images.

As depicted in Figure 1, the UAV is equipped with a visual sensor to capture jet trajectory images from distances of 30 m, 40 m, 50 m, and 60 m away from the fire scene. The frame rate, indicating the number of images captured per second, is set to 30 frames per second. Consequently, the visual sensor captures 30 images per second, and the resulting image size is 1280 pixels wide and 720 pixels high.

### 2.2. Image Preprocessing

#### 2.2.1. Jet Trajectory Datasets

To comprehensively simulate real fire incidents and collect data, the UAV conducted video recordings of jet trajectory from distances of 30 m, 40 m, 50 m, and 60 m away from the fire scene. Equipped with a visual sensor operating at 30 frames per second, each video had a duration of 2 min, ensuring sufficient coverage and detailed capture of the trajectory dynamics. Adhering to the principles of random sampling, 50 frames of jet trajectory images were extracted from each video, ensuring that the data distribution follows an independent distribution pattern. This approach enhances the diversity and representativeness of the dataset, enabling the machine learning models to generalize better across different scenarios and conditions. Moreover, by capturing trajectory images from varying heights, the dataset encompasses a wide range of perspectives, allowing for the analysis of trajectory behavior under different altitude conditions. Additionally, the use of UAVs provides the advantage of flexibility and mobility, enabling the collection of data from different locations and angles, which is crucial for understanding and predicting fire behavior accurately. Furthermore, the high-resolution images captured by the visual sensor ensure detailed observation and analysis of the trajectory morphology, facilitating the development of robust detection and prediction algorithms. Overall, this method of UAV-based data collection combines precision, efficiency, and versatility, making it a valuable tool for fire research and emergency response planning. Image Conversion Algorithm snippets extract frames from a video file, saving them as individual images. It first opens the input video, determines the total number of frames, and calculates the interval for frame extraction. Then, it iterates through each frame, saving frames at specified intervals as separate images until the end of the video is reached. Finally, it releases the video capture resource. Now, we have obtained a training dataset consisting of 200 images of jet trajectory, with the data distribution being independent. Similarly, we perform the same operation to acquire 200 images for the test dataset.

After obtaining the images, the next step involves annotating them, a process vital for training machine learning models. We used the labeling tool named labelimg, a popular annotation tool in the computer vision community, to label the images. This tool enables us to draw bounding boxes around the regions of interest, such as the trajectory of the water jet in our case, and assign corresponding labels. This annotated dataset serves as the foundation for training and evaluating our jet trajectory detection model. Figure 2 illustrates the image processing pipeline applied to the UAV-captured images. In Figure 2a, we present the original image obtained from the UAV sensor, showcasing the raw data captured during flight. Subsequently, in Figure 2b, we demonstrate the image after undergoing label enhancement techniques. These enhancements serve to refine the image quality and emphasize pertinent features, facilitating more accurate analysis and interpretation.

#### 2.2.2. Image Enhancement

Jet trajectory images may suffer from radiometric distortion [31] and geometric distortion [32], where radiometric distortion leads to unrealistic brightness and color in the images, while geometric distortion may cause shifts or deformations in the positions and shapes of the jet trajectory within the images. To address radiometric distortion, radiometric correction methods adjust the exposure and color balance of the images to restore their true radiometric information, while geometric correction methods manipulate the images through rotation, scaling, and translation to correct the positions and shapes of the jet trajectory, making them more accurate and reliable.

As shown in Figure 3, upon receiving the jet trajectory images, the initial step involves resizing each image to dimensions of 1000 and 600 to address potential geometric distortions and ensure uniformity and compatibility across the dataset. Following this, augmentation techniques are applied, where images are randomly flipped horizontally with a probability of 0.5. This flipping process not only enhances the dataset’s diversity but also aids in mitigating geometric distortions, particularly when dealing with varied orientations of jet trajectory in real-world scenarios. Additionally, resizing facilitates efficient processing and analysis of the images, ensuring consistency in feature extraction and model performance across different platforms and computational environments. Figure 4 provides a visual representation of the image-flipping process. The top row displays the original images, while the bottom row exhibits the images after undergoing the flipping transformation. 

Using the specified data augmentation parameters addresses the issue of photometric distortion in jet trajectory images. By adjusting parameters such as brightness (increased by 32 units), contrast (varied within the range of 0.5 to 1.5), saturation (adjusted within the range of 0.5 to 1.5), and hue (increased by 18 units), the photometric characteristics of the images are adjusted to enhance their authenticity and quality. Employing this parameterized data augmentation method is essential, as jet trajectory images are susceptible to photometric distortions, leading to inconsistencies in brightness, contrast, saturation, and hue. Fine-tuning these parameters ensures sharper and more realistic images, thereby enhancing data quality and facilitating the training and performance improvement of subsequent fire detection models. Figure 5 illustrates the visual effects of photometric enhancement applied to the images. The top row depicts the original images, while the bottom row showcases the images after undergoing photometric enhancement. This technique aims to improve image quality by adjusting brightness, contrast, and color balance, thereby enhancing the interpretability and analysis of the images.

### 2.3. Learnable Prompt Vector Generation

#### 2.3.1. Definition of Prompt Learning

Prompt learning is a machine learning paradigm aimed at improving the performance and generalization ability of models by incorporating external prompt information to guide their learning process. These prompts can be specially designed vectors or automatically extracted relevant information from data to assist models in better understanding and processing input data. Prompt learning is widely applied in fields such as computer vision, natural language processing, and reinforcement learning, providing additional context and semantic information to models to tackle complex real-world problems.

The pioneering study to introduce prompt learning to the field of computer vision was CLIP [15]. During training, CLIP uses a contrastive loss to train a shared embedding space for both images and text. In a minibatch of image–text pairs, CLIP maximizes the cosine similarity between each image and its corresponding text while minimizing similarities with all other unmatched texts. This process is repeated for each text as well. Following training, CLIP enables zero-shot image recognition. Let *x* represent the image features from the image encoder, and {*w_i_*}*^N^_i_*_=1_ denote a set of weight vectors from the text encoder, each corresponding to a category (assuming there are *N* categories in total). These weight vectors are derived from prompts, such as “a photo of a {class}”, where the “{class}” token is replaced with the *i*-th class name. Subsequently, the prediction probability is calculated.
(1)$$p(y|x)=\frac{\exp(sim(x,w_{y})/\lambda )}{\sum_{i=1}^{N}\exp(sim(x,w_{i})/\lambda )}$$

*Sim*() denotes cosine similarity, and *λ* is a learned temperature parameter. The aim of CoOp [28] is to address inefficiencies in prompt engineering for enhancing the adaptation of pre-trained vision-language models to downstream applications. Its core concept involves modeling each context token with a continuous vector that can be end-to-end learned from data. Specifically, instead of employing “a photo of a” as the context, CoOp introduces M learnable context vectors, {**u**_1_, **u**_2_, **u**_3_, …, **u**_M_}, each possessing the same dimensionality as the word embeddings. The prompt for the i-th class, denoted by **C**_i_, is then represented as **C**_i_ = {**u**_1_, **u**_2_, **u**_3_, …, **u**_M_, **c**_i_}, where **c**_i_ stands for the word embedding(s) for the class name. These context vectors are shared across all classes. To fine-tune CLIP for a downstream image recognition dataset, a cross-entropy loss is typically employed as the learning objective. Given that the text encoder *E*() is differentiable, gradients can be backpropagated to update the context vectors accordingly. Let *E*() denote the text encoded. Then, the prediction probability is as follows:(2)$$p(y|x)=\frac{\exp(sim(x,E(C_{y}))/\lambda )}{\sum_{i=1}^{N}\exp(sim(x,E(C_{i}))/\lambda )}$$

#### 2.3.2. Offline Module of Learnable Prompt Vectors

To address the jet trajectory image detection problem, we employ the offline module to learn prompt vectors (OMPV), which enhances context vectors with instance-conditional context for improved generalization. The offline module can be trained before the detection and does not take up memory and runtime during the detection. The specific structure is shown in Figure 6. The jet trajectory images are encoded to generate visual feature vectors **I**, which are then inputted into the Meta-Net. The Meta-Net combines these visual feature vectors with token representations of category text, and the concatenated vectors are then fed into the CLIP Text encode to generate text vectors **T**. Finally, a contrastive loss is computed between the visual feature vectors **I** and the text vectors **T**.

Using *φ_θ_*() to denote the Meta-Net parameterized by *θ*, let I represent the image after passing through the encoder. Subsequently, I is fed into *φ_θ_*() to obtain **I***. Next, **I*** is added to the text token **u**_m_ and the context token is now obtained by **u**_m_ (*x*) = **u**_m_ + **I***, where **I*** = *φ_θ_*(**I**) and $m\in \left\{1,\;2,\;\ldots ,M\right\}$. The prompt for the i-th class is thus conditioned on the input, that is, $C_{i}\left(x\right)=\{u_{1}\left(x\right),u_{2}\left(x\right),u_{2}\left(x\right),...,u_{M}\left(x\right),c_{i}\}$. The prediction probability is computed as follows:(3)$$p(y|x)=\frac{\exp(sim(x,E(C_{y}(x))/\lambda )}{\sum_{i=1}^{N}\exp(sim(x,E(C_{i}(x))/\lambda )}$$

Figure 7 illustrates the architecture of the Meta-Net employed in this study, characterized by a two-layer bottleneck structure comprising linear, ReLU, and linear layers. The hidden layer reduces the input dimensionality by a factor of 16×. The Meta-Net receives, as input, the output features generated by the image encoder, facilitating further processing and refinement of the encoded information.

### 2.4. Object Detection Model

Figure 8 depicts the architecture of the object detection model used in this study. The diagram illustrates the sequential stages of the model’s operation. Initially, jet trajectory images undergo data augmentation to enrich the dataset. Subsequently, feature extraction is performed to capture visual characteristics from the augmented images. Following this, proposals are generated using feature pyramids and RoI (Region of Interest) pooling techniques. The RoI Feature Extractor then processes these proposals to extract relevant features. Finally, classification and regression heads are applied, with the classification head incorporating prompts to optimize classification efficiency.

Our methodology builds upon the integration of prompt vectors **v** and classification branches f within the CLIP framework, leveraging the cross-entropy loss function *L_CE_* for effective classification. The core objective is to enhance the model’s ability to classify images by incorporating semantically meaningful prompts alongside traditional classification mechanisms. 

First, we integrate prompt vectors derived from the LPV with the classification branches. This integration is crucial as it allows the model to leverage additional semantic information provided by the prompts during classification. The prompt vectors serve as supplemental cues that guide the classification process towards more accurate and contextually relevant predictions. To facilitate the learning process, we used the cross-entropy loss function *L_CE_* as the optimization objective during training.
(4)$$L_{CE}=-\alpha \sum_{i=1}^{N}v\log(v_{i}^{cls})$$

$v_{i}^{cls}$ is the vector of the cls head, and *α* is a hyperparameter. This loss function quantifies the disparity between the predicted and actual class labels, driving the model to minimize classification errors and improve overall performance. By aligning the model’s predictions with ground truth labels, the cross-entropy loss fosters the development of robust classification capabilities.

## 3. Experiment

In order to verify the proposed PCV method, an experimental facility was constructed at the Ling Tian Co., Ltd. in Xuzhou City, China. Comprehensive Test Site, and experiment results were analyzed to prove the reliability of the proposed characterization methods. The experimental system and experiment result analysis are introduced in the next sections.

### 3.1. Experiment Setup

The experiment was conducted using a comprehensive experimental platform consisting primarily of a UAV system and a firefighting robot, as shown in Figure 9. Auxiliary equipment included control boxes, water cannons, and other supporting devices. The firefighting robot communicated with the UAV through control boxes to enable remote control and manipulation. During the experiment, the UAV was positioned at different distances of water cannon from target points to capture variations in the jet trajectory from different aerial perspectives. Figure 9 vividly illustrates the experimental platform used in this study, showcasing the integration of these devices, which provided reliable technical support for jet trajectory detection. We conducted two comparative experiments, comparing the detection results on the jet trajectory dataset with Faster R-CNN and ATSS, respectively. Then, we conducted five ablation experiments to determine the value of hyperparameter α. 

### 3.2. Parameterization Setup

We collected 50 images at distances of 30 m, 40 m, 50 m, and 60 m from the target point of the fire cracker, respectively, forming a dataset containing 200 jet trajectory images. Each image size was 1280 × 720 pixels.

We used the mmdetection framework, which is a convenient and efficient open-source framework widely used in object detection tasks. The object detection network is based on Faster R-CNN with ResNet-50 as the backbone. We employed stochastic gradient descent (SGD) as the optimizer, with a batch size set to 2. The learning rate was set to 0.02. All experiments were conducted on a single RTX 3060 GPU, with training epochs set to 12. The hyperparameter *α* is set to 1.

Image preprocessing: in terms of image processing, a series of measures were implemented to enhance image quality and accuracy. First, random flipping was applied to increase sample diversity. In our experiments, we set the probability of random flipping during image augmentation to 0.5. This choice is based on the principle of introducing variability while maintaining balance. By setting the probability to 0.5, we ensure that each image has an equal chance of being flipped or not flipped during augmentation. This approach helps to introduce diversity in the dataset, which can improve the robustness and generalization of the model. Additionally, flipping images is a common data augmentation technique used in computer vision tasks to simulate different viewpoints and orientations, thereby enhancing the model’s ability to learn invariant features. Second, geometric corrections, including scaling and padding, were performed to rectify geometric distortions in the images. We resized the images using the Resize operation, setting the width to 1000 pixels and the height to 600 pixels. This choice was driven by both experimental requirements and the input size constraints of our model. Setting “keep_ratio = True” ensured that the aspect ratio of the images remained unchanged, preventing distortion and preserving image accuracy and integrity. Subsequently, we applied the Pad operation to ensure that the image dimensions were divisible by 32. This adjustment was made to meet the input size requirements of our chosen model. Certain convolutional neural network (CNN) architectures necessitate input images with dimensions that are multiples of a specific value. Therefore, padding the images ensured compatibility with the model architecture and improved the efficiency of model training. Lastly, photometric adjustments were conducted to adjust exposure and color balance, thereby restoring the true luminance information of the images and improving the visualization of the jet trajectory. The chosen parameter values are integral to the photometric adjustment process, where they dictate the extent of changes applied to image attributes. In our experiment, setting brightness_delta to 32 allows for a considerable range of brightness adjustments, ensuring a wide spectrum of luminance variations in the augmented dataset. The contrast_range parameter, spanning from 0.5 to 1.5, facilitates both diminishing and amplifying contrast levels, enriching the dataset with diverse contrast variations. Similarly, saturation_range (0.5, 1.5) controls the saturation adjustments, enabling the augmentation of images with varying levels of color intensity. The Hue_delta set at 18 governs the range of hue shifts, introducing subtle variations in color tones across the dataset. These processing steps rendered the images more accurate and reliable, laying a solid foundation for subsequent data analysis and algorithmic applications.

Prediction model: to address the prediction of jet trajectory, we adopted a computer vision approach based on prompt learning. The experimental setup features a Faster R-CNN model configured with a ResNet-50 backbone pre-trained on ImageNet. A Feature Pyramid Network (FPN) with five output stages enhances multi-scale feature extraction using anchor boxes with scales of [8] and ratios of [0.5, 1.0, 2.0]. The Region Proposal Network (RPN) generates object proposals with IoU thresholds (pos_iou_thr = 0.7, neg_iou_thr = 0.3), while RoI sampling ensures balanced selection (pos_fraction = 0.25). During testing, non-maximum suppression is applied to control the number of detections per image (max_per_img = 1000). These parameter configurations aim to optimize object detection performance by balancing computational efficiency and accuracy. This method not only considers the inherent features of the images but also leverages textual prompts to overcome the challenges posed by the limited diversity of jet trajectory scenes and feature scarcity. By incorporating visual and textual multimodal features, we were able to provide a more comprehensive description of jet trajectory characteristics, thereby enhancing the accuracy and robustness of the prediction model.

## 4. Results and Discussion

### 4.1. Experiment Results

The jet trajectory detection network based on prompt-driven computer vision includes key steps such as random flipping, geometric and photometric correction, and learnable prompt vectors. In this study, we conducted a comparative analysis between the improved jet trajectory detection network and traditional methods in both fire detection and jet trajectory detection. To clearly demonstrate the improvements, we extensively compared the prediction results of the new model with Faster R-CNN. During the experiments, we collected 50 jet trajectory images at different distances of the firecracker from the target point (30 m, 40 m, 50 m, and 60 m) using a UAV, resulting in a dataset of 200 annotated images. The experimental results are summarized in Table 1, and the visual detection outcomes are presented in Figure 10: Figure 10(a1–a4) are the results of 30 m, Figure 10(b1–b4) are the results of 40 m, Figure 10(c1–c4) are the results of 0 m, Figure 10(d1–d4) are the results of 60 m.

In the experimental section, we evaluated the performance of the model using two metrics: recall and mean average precision (mAP). Recall measures the proportion of correctly identified positive samples out of the total positive samples, serving as an indicator of the model’s comprehensiveness in detecting targets. It is calculated as follows:(5)$$Recall=\frac{TP}{TP+FN}$$

Here, TP (true positive) represents the number of correctly identified positive samples, while FN (false negative) represents the number of positive samples that were missed by the model.

Mean average precision (mAP) is a widely used performance metric for object detection tasks that considers both precision and recall across different classes. It is calculated by computing the average precision for each class and then taking the harmonic mean of these values. The formula is as follows:(6)$$mAP=\frac{1}{N}\sum_{i=1}^{N}AP_{i}$$

Here, N is the number of classes, and $AP_{i}$ is the average precision for class *i*. These evaluation metrics provide comprehensive insights into the performance of the model at different altitudes, enabling comparisons and optimizations.

From Table 1, when we compare the detection results of our model with other models, it can be observed that the improved model achieved a recall of 0.98, which is 0.25 higher than the Faster R-CNN, and an accuracy of 95.1%, which is 25.4 higher than the Faster R-CNN. Compared with ATSS, our model has a more impressive AP in jet trajectory detection, which is 7.3 higher than ATSS.

Table 2 presents the ablation experiments for each module. Image Enhancement means using the image enhancement when network training. OMPV denotes the offline module of learnable prompt vectors. From Table 2, it can be observed that with the introduction of image enhancement, the recall increased from 0.73 to 0.89 and the accuracy increased from 69.7 to 84.0. The recall increased by 0.16, and the accuracy increased by 14.3. Moreover, the introduction of OMPV further improved the recall to 0.98 and the mean average precision (mAP) to 95.1. With a significant raise already present, the mAP increased by an additional 11.1. This indicates that the introduction of image enhancement increased the probability of Faster R-CNN detection of jet trajectory, while prompts improved the accuracy of jet trajectory classification. In Table 3, we set five different values of the hyperparameter *α*. As the results in Table 3 show, the detection result reaches its best when α is set to 1.0.

### 4.2. Discussion

To investigate the specific effects of Image Enhancement, we conducted a statistical analysis on all annotated boxes (gts) and detected boxes (dets) in the dataset. The experimental results. Figure 11 indicates that after data augmentation, the blue denotes the Faster R-CNN, and the orange denotes the use of image enhancement when training. Yellow indicates using image enhancement and OMPV when training. The number of annotated boxes (gts) remained at 286, while the number of detected boxes increased to 337 after applying Image Enhancement, compared to 315 detected boxes in Faster R-CNN training. The right part of Figure 11 represents the number of predicted anchors. Because of the introduction of prompt learning, the probability of correctly predicted anchors will be higher, and the probability of less effective anchors will be reduced. These less effective anchors will be removed after filtering through the threshold, which results in fewer anchors predicted by Enhancement + Prompt, but the quality will be higher. This demonstrates that Image Enhancement significantly improves the network’s detection capability for targets. 

Figure 12 illustrates the comparison between the detection results obtained from the Faster R-CNN and the improved approach. In subfigures Figure 12(a1,b1,c1), we present the detection results obtained from the Faster R-CNN. Subsequently, in subfigures Figure 12(a2,b2,c2), we showcase the detection results achieved after applying Image Enhancement techniques. This comparison visually demonstrates the effectiveness of Image Enhancement in improving the accuracy and reliability of the detection results. The Faster R-CNN exhibits issues such as missed detections (Figure 12(a1)), inaccurate bounding boxes (Figure 12(b1)), and false detections (Figure 12(c1)). However, the improved method effectively addresses these issues caused by geometric and photometric distortions (Figure 12(a2)), (Figure 12(b2)), and (Figure 12(c2)). This further underscores the effectiveness of Image Enhancement in mitigating geometric and photometric distortions in jet trajectory images due to adverse environmental conditions.

To validate the effectiveness of OMPV, we visualized the scores of the detected boxes. Figure 13 compares the class scores between the Faster R-CNN and PCV detection results. In subfigures a1 and b1, we present the detection results obtained from the Faster R-CNN. Subsequently, in subfigures Figure 13(a2,b2), we showcase the detection results achieved by the proposed approach. This comparison provides a visual representation of the performance improvement achieved by the proposed method in terms of class score accuracy. From the figure, it can be observed that, under the same detected jet trajectory, the classification scores after introducing prompts (Figure 13(a2,b2) are 1.00) are higher than Faster R-CNN ones (Figure 13(a1) is 0.68 and Figure 13(b1) is 0.98). This indicates that, in the case of jet trajectory images with singular features, prompts can learn more effective features by introducing multimodal textual cues, thereby addressing the issue of inaccurate classification.

## 5. Conclusions

The presented novel method for monitoring jet trajectory during the jetting process, integrating UAV imagery with a learnable prompt vector module, has significantly enhanced detection capability and improved monitoring accuracy and stability, achieving a remarkable recall rate of 0.98 and a mean Average Precision (mAP) of 95.1%. Experimental results demonstrate the effectiveness of Image Enhancement techniques in mitigating geometric and photometric distortions, leading to more reliable trajectory monitoring in diverse environmental conditions. The high precision and efficiency achieved hold promise for practical applications in firefighting systems and industrial processes, while future research may explore further optimization and extension of the proposed method to other dynamic monitoring tasks beyond jet trajectory detection.

Future research directions:

Data Diversity and Generalization: Despite the effectiveness of methods such as Image Enhancement and prompt vectors in improving jet trajectory detection, there still exists an issue regarding the diversity and generalization of the dataset. In real-world applications, jet trajectory may be influenced by various environmental conditions, viewing angles, and lighting conditions. Therefore, it is essential for the model to generalize well across diverse scenarios.

Model Interpretability: The interpretability of jet trajectory detection models is crucial in certain critical application scenarios. Users may require insights into the decision-making process of the model to facilitate further analysis and decision-making.

## Figures and Tables

**Figure 1 sensors-24-03553-f001:**
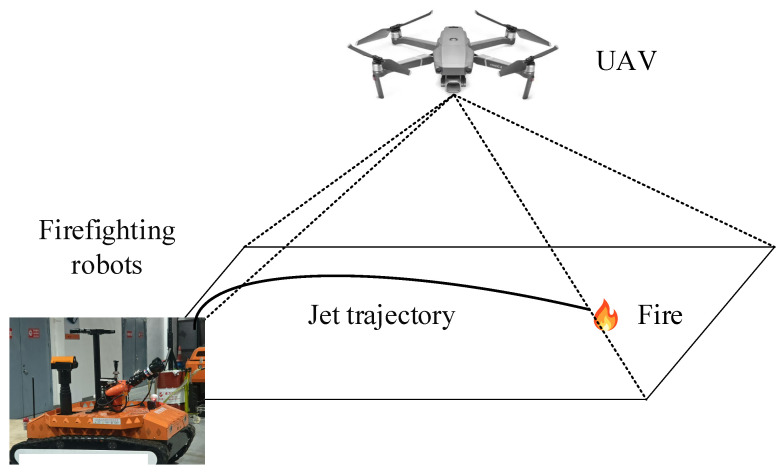
The UAV captures jet trajectory images from a top-down perspective using visual sensors.

**Figure 2 sensors-24-03553-f002:**
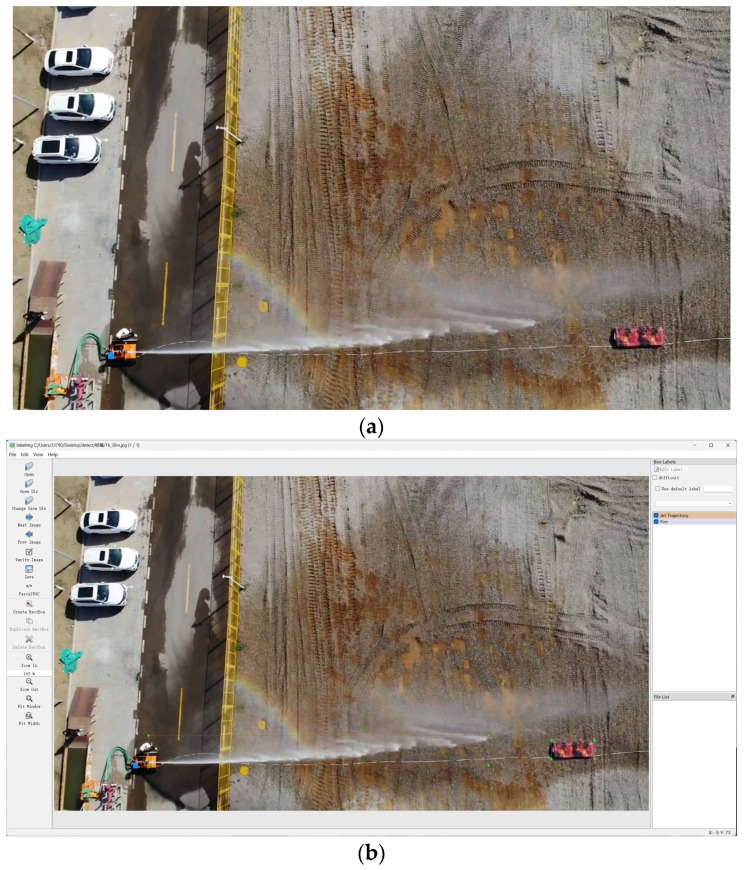
The orange and purple lines represent the labeled boxes, and the green dots represent the four corner points of the boxes. (**a**) is the original image, and (**b**) is the image after the label.

**Figure 3 sensors-24-03553-f003:**
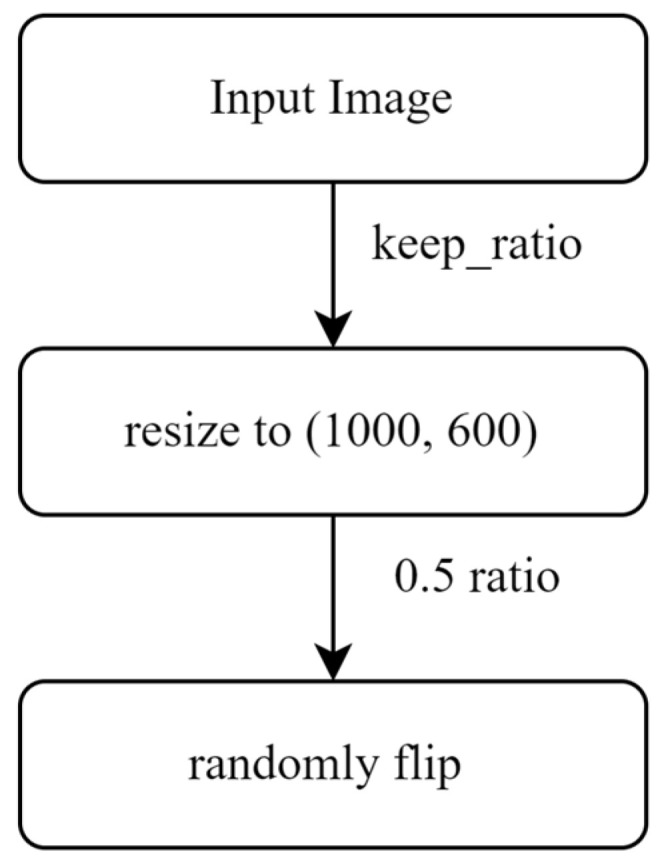
Image flip flow chart.

**Figure 4 sensors-24-03553-f004:**
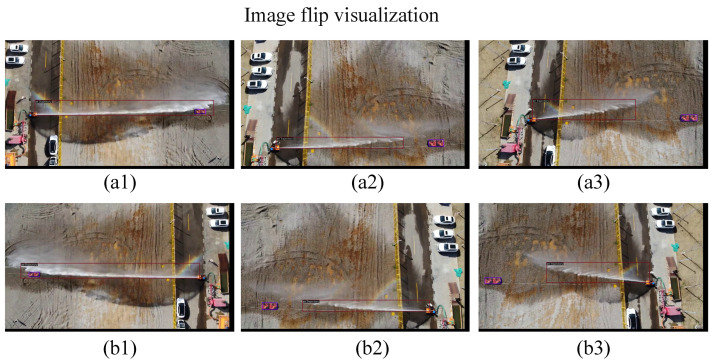
Image flip visualization, the red lines represent the predicted boxes. (**a1**–**a3**) showcase the original images, while (**b1**–**b3**) present the images post-flipping transformation.

**Figure 5 sensors-24-03553-f005:**
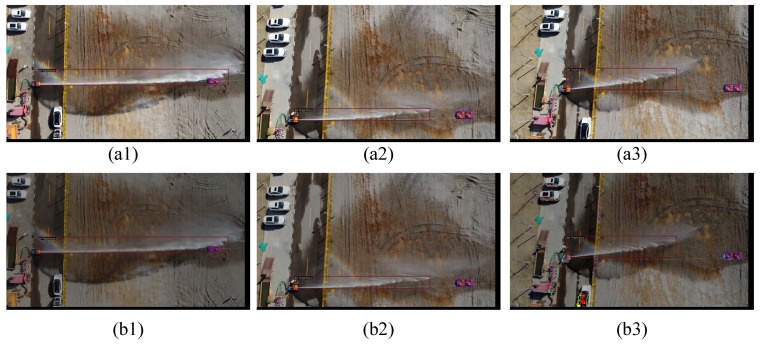
Image photometrics enhance visualization, the red lines represent the predicted boxes. (**a1**–**a3**) present the original images, while (**b1**–**b3**) showcase the images after undergoing photometric enhancement.

**Figure 6 sensors-24-03553-f006:**
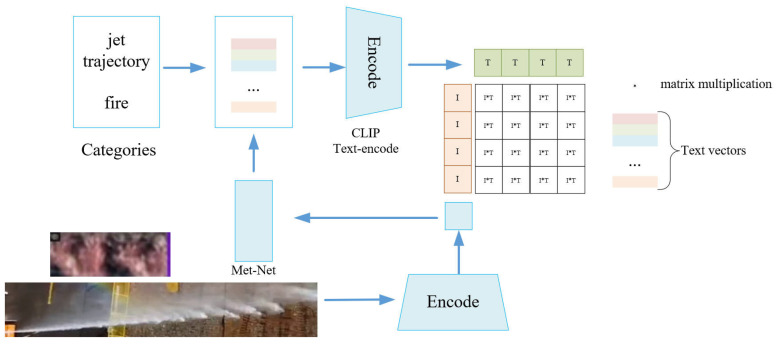
Offline module of learnable prompt vectors.

**Figure 7 sensors-24-03553-f007:**
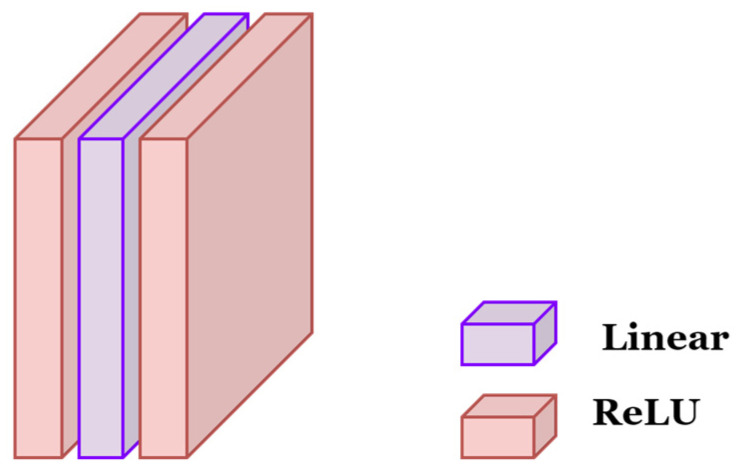
Meta-Net, “Linear” refers to a layer that performs a linear transformation, typically involving matrix multiplication and bias addition. “ReLU” stands for Rectified Linear Unit, which is a popular non-linear activation function that outputs the input value if it is positive and zero otherwise.

**Figure 8 sensors-24-03553-f008:**
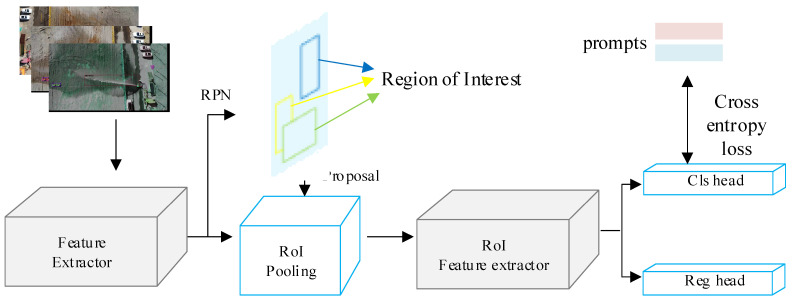
Object detection model. The diagram illustrates the sequential stages of the model’s operation: data augmentation, feature extraction, proposal generation, RoI Feature Extractor, and classification/regression heads. The RoI Feature Extractor processes proposals to extract relevant features, while the classification head incorporates prompts to optimize classification efficiency. The prompts are a supervision of classifications.

**Figure 9 sensors-24-03553-f009:**
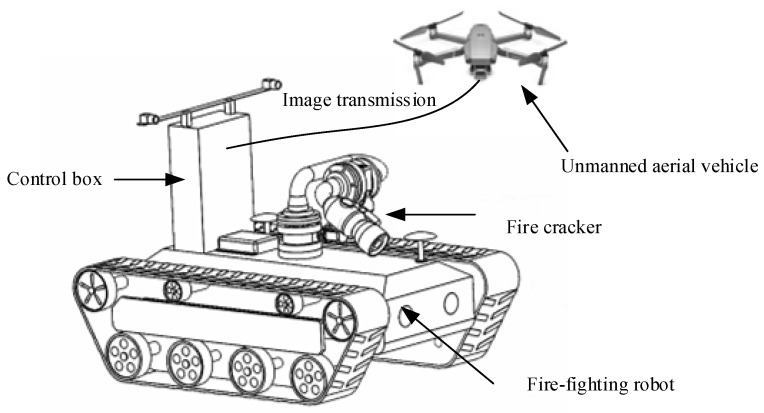
Structural diagram of the experimental platform.

**Figure 10 sensors-24-03553-f010:**
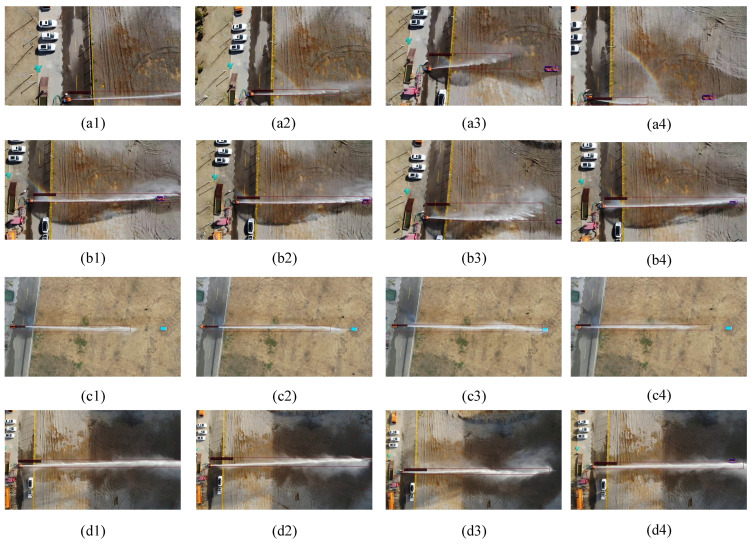
The visualization of jet trajectory detection, the red lines represent the predicted boxes. The visualization illustrates the visual detection outcomes of jet trajectory images collected at various distances (30 m, 40 m, 50 m, and 60 m) from the target point using a UAV during the experiments. The subfigures (**a1**–**a4**) depict the results at 30 m, (**b1**–**b4**) at 40 m, (**c1**–**c4**) at 50 m, and (**d1**–**d4**) at 60 m.

**Figure 11 sensors-24-03553-f011:**
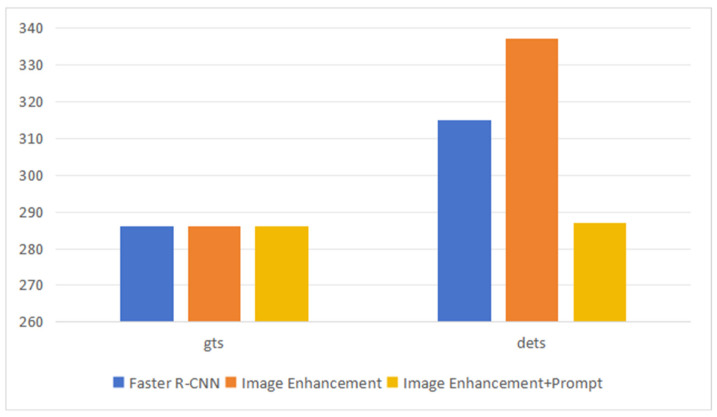
Ablation experiment of gts and dets. gts denotes annotated boxes, and dets denotes detected boxes. Image Enhancement means using the image enhancement when network training. Image Enhancement + Prompt means that we use the image enhancement and offline module of learnable prompt vectors when network training.

**Figure 12 sensors-24-03553-f012:**
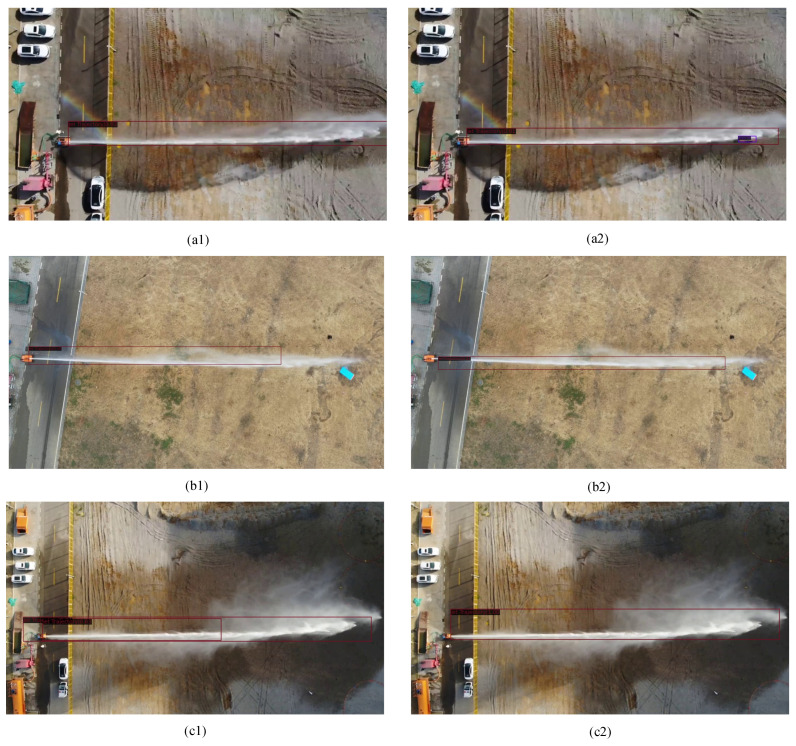
The detection result between Faster R-CNN and improved, the red lines represent the predicted boxes. (**a1**,**b1**,**c1**) present the detection outcomes obtained from the Faster R-CNN, while subfigures (**a2**,**b2**,**c2**) showcase the detection results achieved after applying Image Enhancement techniques.

**Figure 13 sensors-24-03553-f013:**
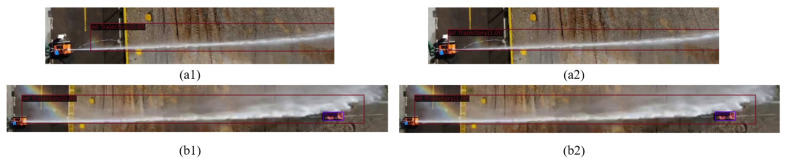
The class score between Faster R-CNN and improved, the red lines represent the predicted boxes. (**a1**,**b1**) display the detection outcomes obtained from the Faster R-CNN, while subfigures (**a2**,**b2**) present the detection results achieved by PCV.

**Table 1 sensors-24-03553-t001:** Comparison between Faster R-CNN and PCV. Faster R-CNN denotes the origin method, while PCV denotes the proposed method. Backbone denotes the backbone network used in model training.

Model	Backbone	Fire	Jet Trajectory	Recall	mAP (%)
ATSS [33]	R-50	99.9	83.5	0.99	91.7
Faster R-CNN	R-50	81.8	57.6	0.73	69.7
PCV	R-50	100	90.3	0.98	95.1

**Table 2 sensors-24-03553-t002:** Ablation of Image Enhancement and OMPV, “√” indicates that the corresponding module was adopted or added during the experiment. Image Enhancement means using the image enhancement when network training. OMPV denotes the offline module of learnable prompt vectors.

Faster R-CNN	Image Enhancement	OMPV	Recall	mAP (%)
√			0.73	69.7
√	√		0.89	84.0
√	√	√	0.98	95.1

**Table 3 sensors-24-03553-t003:** Comparison results (%) to evaluate the effectiveness of different αs on the dataset.

α	Fire	Jet Trajectory	Recall	mAP (%)
1.0	100	90.3	0.98	95.1
0.7	100	90.0	0.97	95.0
0.5	100	94.3	0.97	94.5
0.3	100	79.1	0.90	89.6
0.1	100	89.9	0.98	94.9

## Data Availability

Data are contained within the article.

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
