# Peer review of "Firefighting Water Jet Trajectory Detection from Unmanned Aerial Vehicle Imagery Using Learnable Prompt Vectors"

_sensors, 2024, doi:10.3390/s24113553_

Round 1
Reviewer 1 Report
Comments and Suggestions for Authors
This work presents a novel approach to enhance the quality of images for better monitoring of jet trajectories using UAVs, focusing on addressing image degradation issues through an All-in-one Image Enhancement Network (AIENet) and data augmentation techniques like adjusting brightness, contrast, saturation, and hue to improve image authenticity and quality.
The overall quality is fine. I think it is good to be published after revision.
Detailed comments:
Figure 1 is too simple and hard to understand. It must be a better one.
L110 What is the difference between PCV and Visual Prompt Engineering? I think you need to use the common expression.
[2307.00855] Review of Large Vision Models and Visual Prompt Engineering (arxiv.org)
L134 PCV is an abbreviation in the section title, which should be written as the full name.
L135 I think UAV image capture, image preprocessing UAV image capture, image preprocessing are not good to be a part of PCV.
I recommend you use Visual Prompt Engineering for where you need to write PCV.
L186 Video to Image Conversion Algorithm is too simple to be shown in an science paper.
Each figure in Figure 4,5 has to be added a title.
Comments on the Quality of English Language
fine
Reviewer 2 Report
Comments and Suggestions for Authors
This paper presents a novel and effective methodology for detecting firefighting water jet trajectories using UAV imagery and learnable prompt vectors. Also, it use the offline training method, which reduces the memory and computation load of UAV. The proposed approach offers significant improvements over existing methods and has practical applications in firefighting systems. However, there are some minor points that need to be fixed.
11. In 1st paragraph, the introduction on background of forest fire can be more concise.
22. In part 2.1 UAV image capture, I suggest that you can use UAV equipped with multiple image sensors to get image from different angles. If you did not have that UAV, you can get the image from different angle of fire point instead of just different height.
33. In line 188, the authors said that they use labeling tool, can you provide the name of labeling tool?
44. In figure 4 & 5, can you use the letter (a) , (b), etc to describe the figure instead of just top and bottom?
55. In line 367, can you use full name of SGD, “Stochastic gradient descent”? Because it appears in the paper for the first time.
66. In table 1 and line 524, pls use the specific model name instead of just “ours”.
77. In line 483, the authors specify the gray color presents the result of image enhancement and OMPV method. But in Fig. 11, it is not gray color.
88. In Fig. 11, why the detect points result of Image enhancement + Prompt is worse than that of Faster R-CNN.
